# Niche and Interspecific Association of the Dominant Species during the Invasion of *Alternanthera philoxeroides* in the Yangtze River Basin, China

Qianru Nan , Qing Zhang, Xinghao Li, Danni Zheng, Zhaohua Li and Liya Zhao *

School of Resources and Environmental Science, Hubei University, Wuhan 430062, China
* Correspondence: zhaoly0128@hubu.edu.cn

**Abstract:** The effects of invasive species on the local community (e.g., structure and stability) are highly environmentally dependent. Invasions of amphibious species usually take place in both xeric and humid environments, yet they are relatively poorly understood. In this study, we analyzed the communities that were dominated by *Alternanthera philoxeroides* using ecological niche and interspecific association. A total of 66 species and 67 species were recorded in the xeric environment and humid environment, respectively. In both environments, species in family Gramineae, such as *Echinochloa crusgalli* and *Cynodon dactylon*, exhibited a higher level of importance values and greater ecological niche widths. The interspecific association and stability of the dominant species were weak and the dominant species were relatively independent of each other. In the xeric environment, *A. philoxeroides* was more compatible with *E. crusgalli* and *C. dactylon* in terms of ecological niche requirements and habitat suitability. In humid habitats, *A. philoxeroides* had a greater correlation with *E. crusgalli*, *C. dactylon*, and *Persicaria lapathifolia*, suggesting a higher possibility of concomitant occurrence. Overall, we suggested that during the revegetation after *A. philoxeroides* invasion, *E. crusgalli* and *C. dactylon* can be the alternative plants. Meanwhile, alternative control measures for *A. philoxeroides* invasion in agricultural fields should give more consideration to the use of plants with economic or ecological value.

**Keywords:** *Alternanthera philoxeroides*; invasion; habitat; ecological niche; interspecific association



## 1. Introduction

Recently, the ecological niches, such as the spatial and functional ecological niche, have been one of the key issues for the ecological study of biological invasion [1–6]. By measuring ecological niche characteristics such as the niche width, niche similarity, and niche overlap within and between species, we can quantitatively analyze population relationships, structural characteristics, species diversity, and the resource usage and adaptability in an ecosystem [7].

As one of the most important approaches for estimating ecological niche, interspecific associations can not only be used to quantitatively estimate the interrelationships among species [8], but they are also likely to be an important approach to predict population dynamics and reveal species substitution relationships in community succession [9,10]. The positive or negative associations generally reflect the similarity or dissimilarity of the resource requirements of different species, and, as a consequence, result in the overlap or separation of ecological niches. The combination of ecological niches and interspecific associations, therefore, would provide us with an important reference for ecological conservation and restoration after a biological invasion has taken place [11,12].

For instance, by analyzing the ecological niches and interspecific associations of the dominant plants in the fluctuating-water-level zone of the Three Gorges Reservoir, Zhang et al. found that the interspecific similarities were weak and independent, and proposed a macroscopic layout with *Cynodon dactylon* as pioneer plants and *Setaria viridis*

as companion species to participate in watershed revegetation and reconstruction [13]; Zhang et al. studied the ecological niches and interspecific associations of the woody plants of *Morella rubra* scrub in central Yunnan, indicating that *Morella rubra* was negatively associated with *Vaccinium dunalianum* and *Vaccinium fragile*, and that the ecological niches of these species overlapped to a large extent, which should be paid attention to in the development of the community [14]. Through the study of the niche and interspecific associations of *Pseudoanabaena limnetica*, Ma et al. provided basic data for the study of early warnings regarding phytoplankton community succession and cyanobacteria blooms in large reservoirs in karst areas [15].

Biological invasions are an influential factor in global biodiversity loss, second only to habitat loss and fragmentation as a threat [16–19]. Invasive plants can rapidly encroach on the survival space of native plants in invaded areas and cause the degradation or even disappearance of some native plants, thus, causing significant impacts on the resilience and stability of native ecosystems [20–22].

The invasion of *A. philoxeroides* in China started in the 1950s. This species is a highly competitive amphibious perennial exotic weed and can compete with many crops, such as *Oryza sativa* and *Gossypium* spp., for water, nutrients, and space [23]. In addition, they also reduce crop production through allelopathy [24–26]. Now, *A. philoxeroides* has spread into almost all habitats in the Jianghan Plain, which is located on both sides of the Yangtze River and is an important base for *O. sativa* and *Gossypium* spp. products in China. The invasion of *A. philoxeroides*, therefore, caused a serious series of hazards and potential threats to the ecosystem and agricultural production in China.

Here, we analyzed the ecological niches and interspecific associations to estimate the local community succession status after *A. philoxeroides* has invaded the environment. We aimed to answer the following questions in this study: (1) What are the interspecific associations and interactions between *A. philoxeroides* and major companion species in the habitats invaded by *A. philoxeroides*? (2) Do interspecific associations of dominant species change significantly in communities invaded by *A. philoxeroides* in heterogeneous habitats? (3) Based on the status of ecological niches and interspecific associations of the dominant species during the invasion of *A. philoxeroides* in the Yangtze River basin, we proposed control measures and recommendations.

## 2. Materials and Methods

### 2.1. Study Area

The study was conducted in the lake area of Jianghan Plain (112°35′–113°19′ E, 29°26′–30°12′ N, Hubei Province, China; Figure 1), which is in the subtropical monsoon climate zone, with an annual average temperature of 15.9–16.6 °C, an annual frost-free period of 242–263 d, rainfall of 1100~1300 mm per year, and precipitation mainly concentrated from April to September. The average elevation is approximately 33 m and the soil is mostly sandy loam and clay.

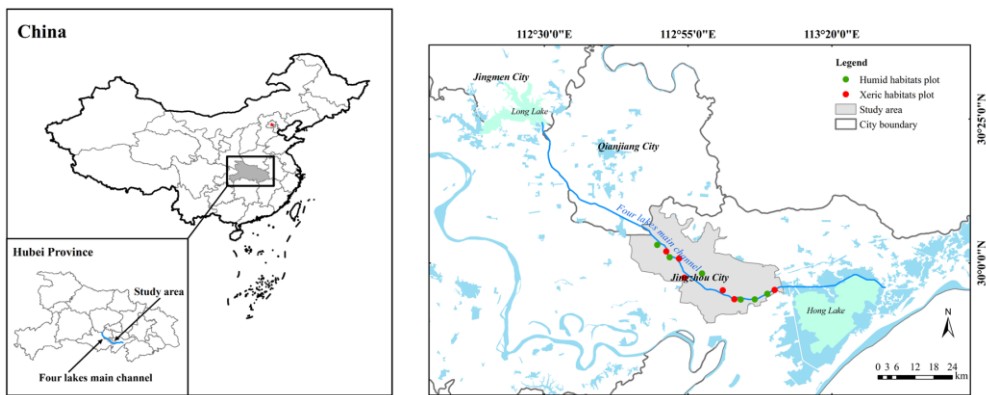

**Figure 1.** Location of sampling sites in the lake area of Jianghan Plain.

### 2.2. Field Survey and the Dominant Species Selection

From 16 July to 3 August 2021, the invasive patch of *A. philoxeroides* was surveyed along the both sides of the Four Lakes Main Channel. Six 10 m × 10 m sample plots were set up in typical xeric (cotton and bean main plantation and abandoned land) and humid (rice plantation, river and lake margins, damp area) environments (Figure 1), and then five 1 m × 1 m quadrats were randomly laid out in each sample plot, resulting in a total of 60 quadrats. Based on the measurement methods of Wu et al., abundance, cover, and height were recorded for each plant species [27]. For all quadrats surveyed, there were 66 species in the xeric environment and 67 species in the humid environment. In this paper, the top 10 species of *A. philoxeroides* invasive communities in the two types of habitats, with a total of 13 major dominant species, were analyzed according to the order of importance values from largest to smallest (Table 1).

**Table 1.** The main species and serial codes of *A. philoxeroides* invasive communities.

| Species Code | Species Name |
| --- | --- |
| AP | *Alternanthera philoxeroides* |
| EC | *Echinochloa crusgalli* |
| CD | *Cynodon dactylon* |
| PT | *Paspalum thunbergia* |
| SV | *Setaria viridis* |
| CR | *Cyperus rotundus* |
| P | *Persicaria lapathifolia* |
| CC | *Conyza canadensis* |
| AS | *Aster subulatus* |
| AA | *Acalypha australis* |
| DS | *Digitaria sanguinalis* |
| Cd | *Cyperus difformis* |
| EP | *Eclipta prostrata* |

### 2.3. Statistical Analysis

We performed statistical analyses of species importance values, ecological niche width, and ecological niche overlap using $\chi^2$ and W tests, the interspecific association index, Pearson and Spearman's rank correlation coefficient tests, etc. on the raw data.

#### 2.3.1. Species Relative Importance Value (IV)

The species importance value is the sum of the various resources available to a species, which collectively characterizes the importance of the plant in the community. We calculated the relative importance value (*IV*) using the following formula: $IV = (RA + RF + RH + RC)/4$ [28].

RA = number of a certain plant species/number of all plant species × 100%

RF = frequency of a certain plant species/frequency of all plant species × 100%

RH = height of a certain plant species/height of all plant species × 100%

RC = coverage of a certain plant species/coverage of all plant species × 100%

where *IV* is the relative importance value, RA is the relative abundance, RF is the relative frequency, RH is the relative height, and RC is the relative coverage.

#### 2.3.2. Niche Width

Niche width is the degree of resource exploitation and utilization of a species. We used the following formulae to assess niche width [29,30]:

$$\text{Levins formula}: \ B_L = \frac{1}{\sum_{j=1}^{r}\left(P_{ij}\right)^2} \tag{1}$$

$$\text{Shannon-Wiener formula}: \ B_S = -\sum_{j=1}^{r} P_{ij}\ln\left(P_{ij}\right) \tag{2}$$

where $B_L$ and $B_S$ are species niche widths, $P_{ij}$ is the proportion of importance values of species $i$ at resource site $j$ to the importance values of species $i$ at all resource sites, and $r$ is the total number of samples, where each sample represents one resource site.

### 2.3.3. Niche Overlap/Niche Similarity

$$\text{Pianka formula}: \ O_{ik} = \frac{\sum_{j=1}^{r} P_{ij} P_{kj}}{\sqrt{\left(\sum_{j=1}^{r} P_{ij}\right)^2 \left(\sum_{j=1}^{r} P_{kj}\right)^2}} \tag{3}$$

$$\text{Schoener formula}: \ C_{ik} = 1 - \frac{1}{2}\sum_{j=1}^{r}\left| P_{ij} - P_{kj}\right| \tag{4}$$

where $C_{ik}$ and $O_{ik}$ are the overlap and similarity indices of resource use curves of species $i$ and species $j$, respectively, with values in the range [0, 1], and larger values indicate higher overlap/similarity of ecological niches; and $P_{ij}$ and $P_{kj}$ are the importance values of species $i$ and $k$ at resource site $j$, respectively [31–33].

### 2.3.4. Overall Association Measurements

According to the variance ratio ($VR$) proposed by Schluter [34], the overall association of the invasive communities of *A. philoxeroides* was determined, and statistical $W$ was used to test the significance of the correlation. The formula is as follows:

$$\begin{aligned}
\delta_T^2 &= \sum_{i=1}^{S} P_i(1 - P_i) \\
S_T^2 &= (1/N)\sum_{j=1}^{N}\left(T_j - t\right)^2 \\
P_i &= n_i/N \\
VR &= S_T^2/\delta_T^2,
\end{aligned} \tag{5}$$

where $S$ is the total number of species, $N$ is the total number of quadrats, $T_j$ is the number of species appearing in quadrat $j$, $n_i$ is the number of quadrats appearing in species $i$, and $t$ is the average number of species in the quadrat.

The variance ratio $VR = 1$ under the assumption of independence, and species show a positive association when $VR > 1$, while $VR < 1$ indicates that the species has a negative association [35]. Then, the formula $W = N \times VR$ was used to test the deviation degree of $VR$ value relative to 1. If the $W$ value fell within the range of $\chi_{0.95}^2(N) < W < \chi_{0.05}^2(N)$, it indicated that there was no significant correlation between species.

### 2.3.5. Interspecific Association Measurements

The $\chi^2$ test, association coefficient ($AC$), percentage of co-occurrence ($PC$), Ochiai index ($OI$), and Dice index ($DI$) were used to test interspecific associations. We established a $2 \times 2$ contingency table to count the values of a, b, c, and d of species pairs, where a is the number of plots where species A and species B occur together, b is the number of plots where species A and species B do not occur together, c is the number of plots where only species A occurs, and d is the number of plots where only species B occur. To avoid the denominator value being 0, the value of b and d being 0 is weighted to 1 [36].

(1) $\chi^2$ test

The $\chi^2$ test is a qualitative judgment method that uses the occurrence of each species in the sample as a criterion for judging the interspecific association, which is a simple method that can objectively and accurately reflect the relationship among species [36].

The $\chi^2$ value was calculated using Yates's continuity correction formula [37]:

$$\chi^2 = \frac{N\left[(\mid ad - bc \mid) - \frac{1}{2}N\right]}{(a + b)(c + d)(a + c)(b + d)} \tag{6}$$

where $N$ is the total sample number. When $\chi^2 < 3.841$, the association between species pairs is independent; when $\chi^2 > 6.635$, there is a significant ecological association between species pairs; when $3.841 < \chi^2 < 6.635$ and $ad > bc$, there is some positive association between species pairs, and the opposite is a negative association.

(2) Association coefficient ($AC$)

The association coefficient ($AC$) was used to test the result of the $\chi^2$ test and show the degree of interspecific association. It was calculated as follows [38]:

$$AC = \begin{cases} \frac{ad-bc}{(a+b)(b+d)}, & ad - bc \geq 0 \\ \frac{ad-bc}{(a+b)(a+c)}, & ad - bc < 0 \ and \ d - a \geq 0 \\ \frac{ad-bc}{(a+b)(a+c)}, & ad - bc < 0 \ and \ d - a < 0 \end{cases} \tag{7}$$

The $AC$ range was [–1, 1]; when $AC$ = 1, it indicates the strongest positive interspecific association between species, and $AC$ = $-1$ means that the species pair has the strongest negative interspecific association, while $AC$ = 0 means the species pair is completely independent.

(3) Percentage of co-occurrence ($PC$)

The percentage of co-occurrence ($PC$) was used to measure the degree of positive association. The formula is as follows [39]:

$$PC = \frac{a}{a + b + c} \tag{8}$$

The range of value of $PC$ is [0, 1]. The closer the $PC$ is to 1, the stronger the positive interspecific association.

(4) Ochiai/Dice index ($OI/DI$) [39]

The $OI/DI$ index indicates the probability of concomitant occurrence of species pairs and the degree of association, and takes values in the range of [0, 1], with the following formulae:

$$OI = \frac{a}{\sqrt{a + b} \cdot \sqrt{a + c}} \tag{9}$$

$$DI = \frac{2a}{2a + b + c} \tag{10}$$

(5) Pearson and Spearman's rank correlation coefficients [40]

Pearson and Spearman's rank correlation coefficients were used to correlate the interspecific relationships of the communities based on the importance values with the following formulae:

$$r_p(i,k) = \frac{\sum_{j=1}^{N}\left(x_{ij} - \bar{x}_i\right)\left(x_{kj} - \bar{x}_k\right)}{\sqrt{\sum_{j=1}^{N}\left(x_{ij} - \bar{x}_i\right)^2 \sum_{j=1}^{N}\left(x_{kj} - \bar{x}_k\right)^2}} \tag{11}$$

$$r_s(i,k) = 1 - \frac{6\sum_{j=1}^{N}\left(x_{ij} - \bar{x}_i\right)^2\left(x_{kj} - \bar{x}_k\right)^2}{N^3 - N} \tag{12}$$

where $N$ is the total number of samples, and $x_{ij}$ and $x_{kj}$ are the rank (importance value) of species $i$ and species $k$ in sample $j$, respectively.

### 2.3.6. Data Processing

Data analyses were performed using Excel 2019, R 3.5.2 (Lucent Technologies, Murray Hill, NJ, USA), and SPSS 21.0 (SPSS Inc., Chicago, IL, USA). Data were processed using Excel 2019 and SPSS 21.0 (SPSS Inc., Chicago, IL, USA), and semi-matrices were plotted using the ggplot2 package in R 3.5.2 (Lucent Technologies, Murray Hill, NJ, USA). Interspecific associations, niche width, and niche overlap were determined using the spaa package in R 3.5.2 (Lucent Technologies, Murray Hill, NJ, USA).

## 3. Results

### 3.1. Dominant Species and the Importance Values

Xeric environment: There were 66 species in 60 genera and 29 families. *A. philoxeroides*, *Echinochloa crusgalli*, *C. dactylon*, *Paspalum thunbergii*, *S. viridis*, *Cyperus rotundus*, *P. lapathifolia*, *Conyza canadensis*, *Aster subulatus*, and *Acalypha australis* were the dominant species, which accounted for 65.04% of the total importance values.

Humid environment: There were 67 species in 58 genera and 21 families. *A. philoxeroides*, *C. dactylon*, *E. crusgalli*, *Digitaria sanguinalis*, *P. lapathifolia*, *P. thunbergii*, *Cyperus difformis*, *C. canadensis*, *Eclipta prostrata*, and *C. rotundus* were the dominant species, which accounted for 66.09% of the total importance values.

The importance values of *A. philoxeroides*, *E. crusgalli*, and *C. dactylon* ranked in the top three among the dominant species in both habitats. Although the plant families Amaranthaceae, Poaceae, and Asteraceae were dominant, their importance values differed between the xeric and humid environment. For instance, the total importance values of Amaranthaceae and Asteraceae in the xeric environment were 12.24% and 11.79%, respectively, which were higher than those in the humid environment (Table 2).

**Table 2.** The importance value and niche width of dominant plants.

| Species | Family | Xeric Habitats | | | Humid Habitats | | |
|---|---|---|---|---|---|---|---|
| | | Niche Width | | Importance Value (%) | Niche Width | | Importance Value (%) |
| | | Levins Index (BL) | Shannon–Wiener Index (BS) | | Levins Index (BL) | Shannon–Wiener Index (BS) | |
| *Alternanthera philoxeroides* | Amaranthaceae | 26.20 | 3.33 | 25.77 | 25.84 | 3.32 | 23.52 |
| *Echinochloa crusgalli* | Gramineae | 16.13 | 2.90 | 8.16 | 10.61 | 2.57 | 10.16 |
| *Cynodon dactylon* | Gramineae | 10.46 | 2.43 | 6.93 | 10.84 | 2.46 | 13.99 |
| *Paspalum thunbergii* | Gramineae | 3.99 | 1.55 | 6.28 | 4.41 | 1.55 | 3.34 |
| *Setaria viridis* | Gramineae | 8.46 | 2.25 | 4.22 | — | — | 1.11 |
| *Persicaria lapathifolia* | Polygonaceae | 6.70 | 1.98 | 3.15 | 10.93 | 2.51 | 4.24 |
| * *Conyza canadensis* | Asteraceae | 6.95 | 2.05 | 2.90 | 6.61 | 1.92 | 2.94 |
| * *Aster subulatus* | Asteraceae | 6.08 | 1.87 | 2.51 | — | — | 1.17 |
| * *Cyperus rotundus* | Cyperaceae | 8.18 | 2.21 | 3.16 | 5.60 | 1.76 | 2.16 |
| *Digitaria sanguinalis* | Gramineae | — | — | 1.81 | 5.25 | 1.95 | 4.59 |
| *Cyperus difformis* | Cyperaceae | — | — | — | 6.04 | 2.02 | 3.18 |
| *Acalypha australis* | Euphorbiaceae | 6.21 | 1.96 | 1.95 | — | — | — |
| * *Eclipta prostrata* | Asteraceae | — | — | 1.47 | 7.71 | 2.06 | 2.21 |

* Alien plant; —: The plant did not appear in the habitat.

### 3.2. Analysis of Niche Characteristics

The ecological niche width of the dominant species during the invasion in the xeric environment ranged from 3.99 to 26.20 for the Levins index ($B_L$) and from 1.55 to 3.33 for the Shannon–Wiener index ($B_S$). This was similar with that in the humid environment ($B_L$: 4.41–25.84, $B_S$: 1.55–3.32).

We also observed substantial variation in the niche width between the different dominant species. In the xeric environment, *A. philoxeroides* occupied a greater niche width (i.e., $B_L$: 26.20, $B_S$: 3.33). Similar trends were also observed for $B_S$. In the humid environment, the $B_L$ and $B_S$ of *A. philoxeroides* were also greatest (i.e., 25.84 and 3.32).

The ecological niche similarity index ($C_{ik}$) of 45 species pairs in the xeric environment ranged from 0.00 to 0.60, with a mean value of 0.23 (Table 3). There was only one species pair with $C_{ik} \geq 0.50$ (i.e., *A. philoxeroides*–*E. crusgalli*: 0.60), indicating that *A. philoxeroides* and *E. crusgalli* have similar resource or environmental requirements, and the field survey also found that these two species were concentrated in certain resource niches. The $C_{ik}$ of 27 species pairs (60% of the total) ranged from 0.20 to 0.50. The $C_{ik}$ of 17 species pairs (37.78% of the total) were in the range of from 0.00 to 0.20.

In the humid environment, the highest $C_{ik}$ value (0.50) was found for species pair *A. philoxeroides*–*E. crusgalli*. The $C_{ik}$ value of each one of the remaining species pairs was less than 0.50, with a mean value of 0.24, indicating that the interspecific resource requirements in the humid habitats varied greatly.

**Table 3.** Schoener overlap index (Cik) (diagonal to the left) and Pianka overlap index (Oik) (diagonal to the right) of main plant species in the xeric habitats.

|      | AP   | EC   | CD   | PT   | SV   | CR   | P    | CC   | AS   | AA   |
|------|------|------|------|------|------|------|------|------|------|------|
| AP   |      | 0.71 | 0.45 | 0.31 | 0.46 | 0.50 | 0.32 | 0.48 | 0.38 | 0.48 |
| EC   | 0.60 |      | 0.36 | 0.00 | 0.40 | 0.12 | 0.44 | 0.53 | 0.39 | 0.46 |
| CD   | 0.33 | 0.32 |      | 0.28 | 0.31 | 0.32 | 0.38 | 0.50 | 0.16 | 0.23 |
| PT   | 0.17 | 0.00 | 0.27 |      | 0.01 | 0.17 | 0.08 | 0.02 | 0.02 | 0.02 |
| SV   | 0.30 | 0.38 | 0.31 | 0.03 |      | 0.46 | 0.17 | 0.32 | 0.46 | 0.07 |
| CR   | 0.33 | 0.13 | 0.36 | 0.24 | 0.28 |      | 0.04 | 0.14 | 0.21 | 0.32 |
| P    | 0.21 | 0.32 | 0.30 | 0.12 | 0.19 | 0.05 |      | 0.19 | 0.28 | 0.13 |
| CC   | 0.29 | 0.34 | 0.44 | 0.04 | 0.33 | 0.17 | 0.15 |      | 0.11 | 0.10 |
| AS   | 0.20 | 0.27 | 0.19 | 0.04 | 0.41 | 0.23 | 0.26 | 0.15 |      | 0.29 |
| AA   | 0.27 | 0.28 | 0.23 | 0.04 | 0.13 | 0.35 | 0.14 | 0.15 | 0.33 |      |

Species codes are the same as in Table 1.

From the overall distribution of similarity values (Figure 2a), the ecological niche similarity distribution maps of the xeric and humid environments basically overlapped, and both were mainly distributed in the [0, 0.4] interval, indicating that most species in the habitats had relatively different resource requirements.

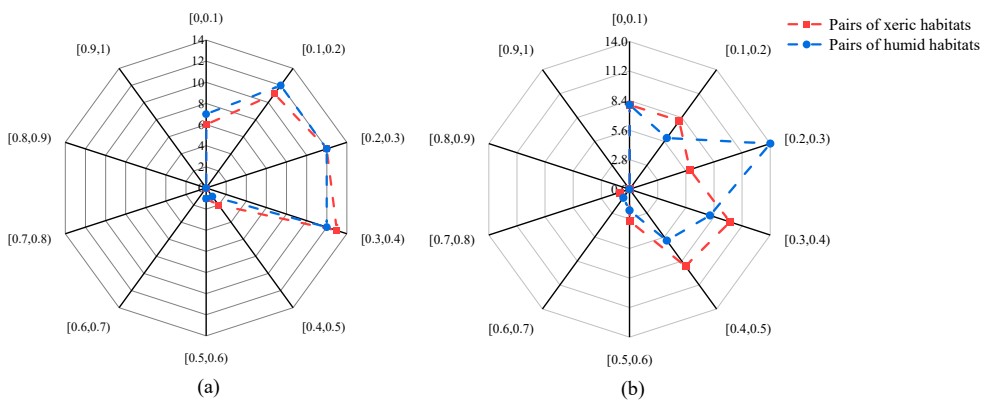

**Figure 2.** Distribution map of the ecological niche similarity (Cik) and niche overlap (Oik) values of invasion communities of *A. philoxeroides*. (**a**) Distribution map of the niche similarity values. (**b**) Distribution map of the niche overlap values. The red and blue lines represent the exponential ranges of xeric and humid habitats, respectively. Red and blue dots represent the number of species pairs in xeric and humid habitats at the corresponding index ranges, respectively.

The niche overlap index ($O_{ik}$) of the communities in the xeric environment ranged from 0.00 to 0.71 (Table 4). Among them, there were four pairs with $O_{ik} \geq 0.50$: *A. philoxeroides–E. crusgalli* (0.71), *C. canadensis–E. crusgalli* (0.53), *A. philoxeroides–C. rotundus* (0.50), and *C. dactylon–C. canadensis* (0.50), which accounted for 8.89% of the total number of pairs. The lowest $O_{ik}$ was found in the pairs of *E. crusgalli–P. thunbergii* (0.00) and *S. viridis–P. thunbergii* (0.01), indicating that the niche overlap between these pairs was extremely low or did not occur, and the species existed independently of each other.

In the humid environment, the overlap index ($O_{ik}$) ranged from 0.00 to 0.66. The $O_{ik}$ of tree species pairs was more than 0.50, and they were *A. philoxeroides–E. crusgalli* (0.66), *C. rotundus–C. dactylon* (0.53), and *P. lapathifolia–P. thunbergii* (0.50), accounting for 6.67% of the total number of species pairs.

**Table 4.** Schoener overlap index (Cik) (diagonal to the left) and Pianka overlap index (Oik) (diagonal to the right) of main plant species in the humid habitats.

|    | AP | CD | EC | DS | P | PT | Cd | CC | EP | CR |
|----|----|----|----|----|----|----|----|----|----|----|
| AP |    | 0.41 | 0.66 | 0.46 | 0.48 | 0.32 | 0.46 | 0.39 | 0.34 | 0.28 |
| CD | 0.31 |    | 0.21 | 0.23 | 0.39 | 0.21 | 0.14 | 0.37 | 0.24 | 0.53 |
| EC | 0.50 | 0.24 |    | 0.09 | 0.25 | 0.00 | 0.12 | 0.16 | 0.32 | 0.09 |
| DS | 0.30 | 0.35 | 0.15 |    | 0.12 | 0.04 | 0.04 | 0.26 | 0.20 | 0.24 |
| P  | 0.39 | 0.39 | 0.25 | 0.19 |    | 0.50 | 0.27 | 0.23 | 0.48 | 0.26 |
| PT | 0.15 | 0.18 | 0.00 | 0.05 | 0.31 |    | 0.21 | 0.11 | 0.08 | 0.00 |
| Cd | 0.29 | 0.22 | 0.18 | 0.08 | 0.30 | 0.27 |    | 0.15 | 0.35 | 0.06 |
| CC | 0.20 | 0.27 | 0.13 | 0.32 | 0.19 | 0.13 | 0.17 |    | 0.35 | 0.41 |
| EP | 0.19 | 0.21 | 0.26 | 0.22 | 0.37 | 0.09 | 0.42 | 0.36 |    | 0.26 |
| CR | 0.15 | 0.38 | 0.08 | 0.25 | 0.19 | 0.00 | 0.08 | 0.37 | 0.28 |    |

Species codes are the same as in Table 1.

### 3.3. Overall Association

The overall association of *A. philoxeroides* between xeric and humid environments was greater than 1, indicating a positive association between the major species and *A. philoxeroides*. The test statistic (W = 37.97) for the xeric habitat was within the interval of $\chi^2_{0.95}(N) = 18.49$ and $\chi^2_{0.05}(N) = 43.77$, indicating that the dominant species in the xeric environment showed an insignificant positive association with *A. philoxeroides*, i.e., there was some association between the populations, but no stable interaction was formed (Table 5).

**Table 5.** Overall interspecific associations among dominant plant populations in *A. philoxeroides* invasive communities.

| Type of Habitat | Variance Ratio | Test Statistic W | $\chi^2$ Critical Value | Results |
|-----------------|----------------|------------------|-------------------------|---------|
| Xeric habitats  | 1.27 | 37.97 | (18.49, 43.77) | No significant positive association |
| Humid habitats  | 1.49 | 44.65 | (18.49, 43.77) | Significant positive association |

In contrast, the test statistic of the humid habitat (W = 44.65) was not within the $\chi^2$ critical value interval, indicating that there was a significant positive association among the community in humid habitats.

### 3.4. Interspecific Association

#### 3.4.1. $\chi^2$ Test

In the xeric environment, the $\chi^2$ statistic of the species pair *E. crusgalli–P. thunbergia* (13.581) was greater than 6.635, which indicated that the association was significant, and the species pair *E. crusgalli–C. rotundus* (4.464) was greater than 3.841, which showed a certain association. The $\chi^2$ values of the remaining species pairs were not significant ($\chi^2 < 3.841$). Among the humid environment, only the $\chi^2$ statistic of the species pair *E. crusgalli–P. thunbergii* (5.321) was greater than 3.841, showing some negative association. Overall, the $\chi^2$ test suggested that the association of the dominant species in these two different environments was weak (Tables 6 and 7).

**Table 6.** $\chi^2$ statistical data matrix of populations in xeric habitats.

| AP | | | | | | | | | |
|---|---|---|---|---|---|---|---|---|---|
| 0.020 | EC | | | | | | | | |
| 0.240 | 0.103 | CD | | | | | | | |
| 0.026 | 13.581 | 0.687 | PT | | | | | | |
| 0.054 | 1.607 | 0.831 | 2.109 | SV | | | | | |
| 0.054 | 4.464 | 0.831 | 2.109 | 0.469 | CR | | | | |
| 0.002 | 0.657 | 0.001 | 0.011 | 0.021 | 1.044 | P | | | |
| 0.020 | 1.088 | 1.655 | 0.089 | 0.179 | 0.179 | 0.008 | CC | | |
| 0.003 | 0.319 | 0.165 | 0.012 | 1.141 | 0.023 | 0.382 | 0.142 | AS | |
| 0.002 | 0.008 | 0.001 | 0.011 | 0.021 | 0.533 | 0.117 | 0.008 | 2.542 | AA |

Species codes are the same as in Table 1.

**Table 7.** $\chi^2$ statistical data matrix of populations in humid habitats.

| AP | | | | | | | | | |
|---|---|---|---|---|---|---|---|---|---|
| 0.240 | CD | | | | | | | | |
| 0.240 | 0.010 | EC | | | | | | | |
| 0.020 | 0.233 | 0.233 | DS | | | | | | |
| 0.331 | 0.102 | 1.121 | 0.057 | P | | | | | |
| 0.080 | 0.109 | 5.321 | 0.000 | 1.313 | PT | | | | |
| 0.020 | 0.103 | 0.103 | 1.088 | 0.057 | 1.143 | Cd | | | |
| 0.003 | 0.165 | 0.165 | 1.739 | 0.041 | 0.149 | 0.142 | CC | | |
| 0.002 | 0.001 | 2.685 | 0.008 | 0.403 | 0.034 | 3.580 | 0.382 | EP | |
| 0.026 | 3.063 | 0.008 | 0.486 | 0.410 | 0.375 | 0.089 | 1.409 | 0.011 | CR |

Species codes are the same as in Table 1.

### 3.4.2. Association Coefficient (*AC*) and Percentage of Co-Occurrence (*PC*)

The *AC* values in xeric environments ranged from −1.0 to 0.5, with 20 positively associated species pairs and 25 negatively associated species pairs (Figure 3). The *AC* values of 56% negatively associated species pairs ranged from −0.3 to 0, which suggested a weak negative association. The *AC* values of species pairs *E. crusgalli*–*P. thunbergii* (−1.00), *P. thunbergia*–*S. viridis* (−1.00), and *C. rotundus*–*P. lapathifolia* (−0.63) were lower than −0.50, which showed a significant negative association. The *AC* values of *E. crusgalli*–*C. dactylon* and *E. crusgalli*–*A. australis* were close to 0, indicating that *E. crusgalli* was relatively independent of *C. dactylon* and *A. australis*. The *AC* values of almost all (nearly 90%) positively associated species pairs, except for *P. thunbergia*–*C. rotundus* (0.50), ranged from 0 to 0.3, indicating weak species associations in the xeric environment.

The *AC* values in the humid environments ranged from −0.38 to 0.43, with 22 positively associated species pairs and 23 negatively associated species pairs, indicating the interspecific association was weak and the species were relatively independent. This was consistent with the results of the $\chi^2$ test. The species pairs, except for the weak positive association of *A. philoxeroides* with *E. crusgalli*, were negatively associated, indicating that *A. philoxeroides* is highly adaptive and rapidly invasive, which makes it difficult for other species to adapt to grow alongside it, thus, leading to competitive exclusion [41,42].

The co-occurrence percentage ranged from 0 to 0.68 in the xeric environment (Figure 4). Only the PC value of species pair *A. philoxeroides*–*E. crusgalli* was more than 0.5, suggesting that they were significantly or highly significantly associated. The *PC* values of 13 species pairs (28.89%) ranged from 0.3 to 0.5. The *PC* values of *A. philoxeroides*–*C. dactylon* and *E. crusgalli*–*S. viridis* were relatively higher than others, indicating higher levels of interspecific association. The *PC* values of 24 species pairs (53.33%) ranged from 0.1 to 0.3. The *PC* values of seven species pairs ranged from 0 to 0.1, indicating that the interrelationship between species was looser and the co-occurrence was weak.

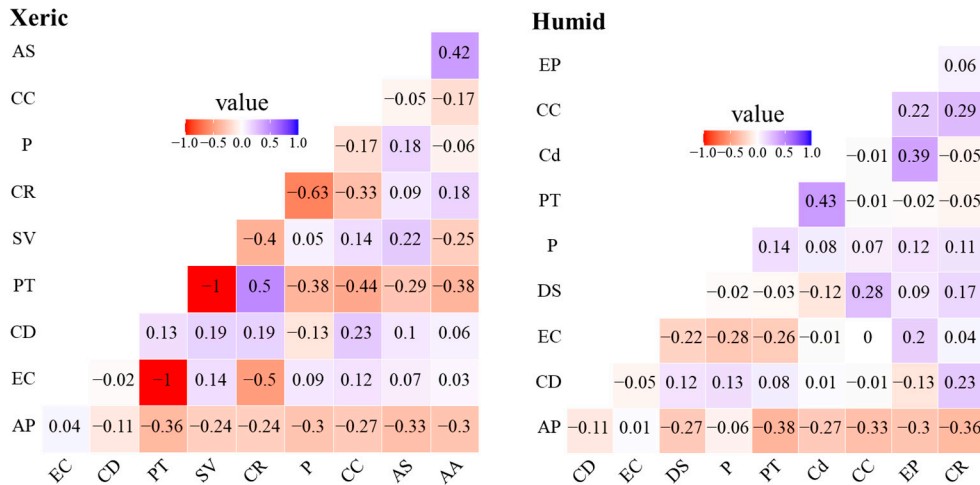

**Figure 3.** Semi-matrix diagram of association coefficients of dominant plant species in *A. philoxeroides* invasive communities. Species codes are the same as in Table 1.

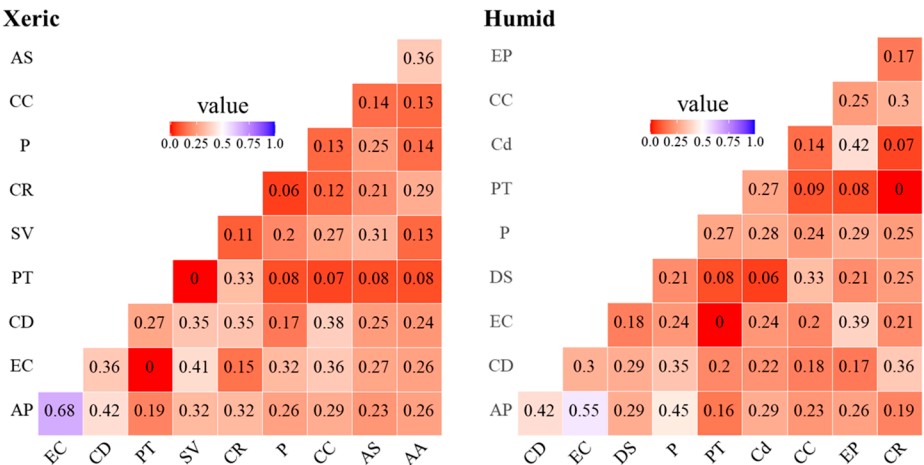

**Figure 4.** Semi-matrix diagram of percentage of co-occurrence of dominant species in *A. philoxeroides* invasive communities. Species codes are the same as in Table 1.

In the humid environment, the *PC* values ranged from 0 to 0.55. Only the *PC* value of *A. philoxeroides*–*E. crusgalli* was greater than 0.5, indicating that *A. philoxeroides* and *E. crusgalli* are strongly related to each other. This may be related to the ecological characteristics and adaptability of the two species to the living environment. The *PC* values of nine species pairs (20.00%) ranged from 0.3 to 0.5. The PC values of 28 species pairs (62.22%) ranged from 0.1 to 0.3. The *PC* values of seven species pairs were lower than 0.1.

### 3.4.3. Ochiai/Dice Index (OI/DI)

The OI values in the xeric environment ranged from 0 to 0.82 (Figure 5). The percentages of the species pairs with OI values ranging from 0 to 0.2, from 0.2 to 0.4, from 0.4 to 0.6, and from 0.6 to 1 were 15.56%, 26.67%, 51.11%, and 6.67%, respectively. The OI values of three species pairs (i.e., *A. philoxeroides*–*E. crusgalli*, *A. philoxeroides*–*C. dactylon*, and *E. crusgalli*–*S. viridis*,) were greater than 0.6, indicating that *A. philoxeroides* was more likely to occur in association with *E. crusgalli* and *C. dactylon*. The OI values of seven species pairs were less than 0.2. The remaining 77.78% of the species pairs had OI values ranging from 0.2 to 0.6.

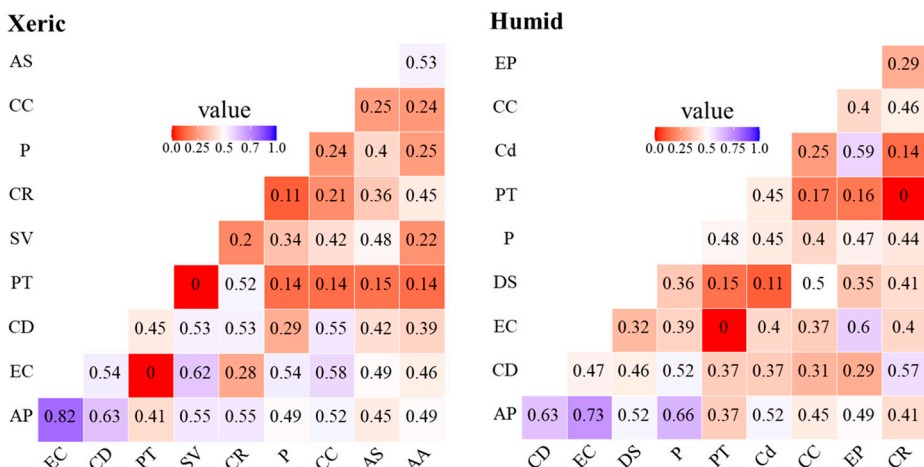

**Figure 5.** Semi-matrix diagram of Ochiai index of dominant species in *A. philoxeroides* invasive communities. Species codes are the same as in Table 1.

In the humid environment, the OI values ranged from 0 to 0.73. The percentages of the species pairs with OI values ranging from 0 to 0.2, from 0.2 to 0.4, from 0.4 to 0.6, and from 0.6 to 1 were 15.56%, 26.67%, 48.89%, and 8.89%, respectively. The species pairs with OI values greater than 0.6 included *A. philoxeroides–E. crusgalli*, *A. philoxeroides–P. lapathifolia*, and *A. philoxeroides–C. dactylon*, indicating that *A. philoxeroides* had a stronger correlation with *E. crusgalli*, *C. dactylon*, and *P. lapathifolia*. The OI values of seven species pairs were less than 0.2. The DI and OI values of *A. philoxeroides* were generally consistent with each other in the different environments (Figure 6).

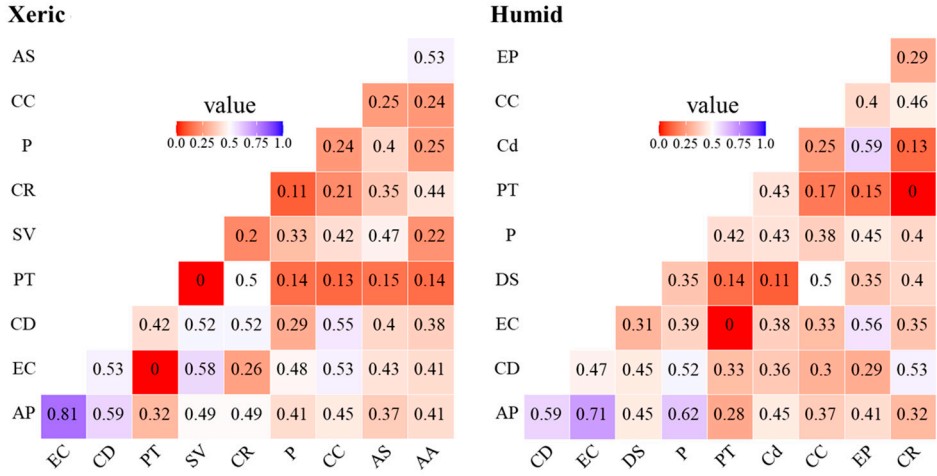

**Figure 6.** Semi-matrix diagram of Dice index of dominant species in *A. philoxeroides* invasive communities. Species codes are the same as in Table 1.

### 3.4.4. Pearson and Spearman's Rank Correlation Coefficients

The Pearson correlation coefficient analysis (Tables 8 and 9) revealed that three species pairs in the xeric environment were significantly negatively correlated. They were *A. philoxeroides–P. lapathifolia*, *E. crusgalli–P. thunbergii*, and *E. crusgalli–C. rotundus*. In the humid environment, two species pairs were significantly positively correlated. They were *C. dactylon–C. rotundus* and *P. lapathifolia–P. thunbergia*. Two species pairs, *A. philoxeroides–C. dactylon* and *A. philoxeroides–E. prostrata*, showed significant negative correlations.

**Table 8.** Pearson (diagonal to the left) and Spearman's (diagonal to the right) correlation half-matrix table of main plant species in the xeric habitats.

| | AP | EC | CD | PT | SV | CR | P | CC | AS | AA |
|---|---|---|---|---|---|---|---|---|---|---|
| AP | | 0.102 | −0.398 ** | −0.222 | −0.249 | −0.032 | −0.395 * | 0.007 | −0.150 | 0.094 |
| EC | 0.086 | | −0.115 | −0.610 ** | 0.119 | −0.527 ** | 0.224 | 0.093 | 0.187 | 0.050 |
| CD | −0.356 | −0.130 | | 0.242 | 0.168 | 0.165 | −0.007 | 0.364 * | 0.010 | −0.002 |
| PT | −0.102 | −0.422 * | 0.085 | | −0.273 | 0.285 | −0.111 | −0.181 | −0.131 | −0.156 |
| SV | −0.121 | 0.011 | −0.002 | −0.238 | | −0.020 | −0.027 | 0.192 | 0.382 * | −0.126 |
| CR | 0.039 | −0.450 * | 0.017 | −0.020 | 0.247 | | −0.298 | −0.128 | 0.063 | 0.233 |
| P | −0.390 * | 0.152 | 0.144 | −0.118 | −0.107 | −0.272 | | −0.062 | 0.173 | −0.061 |
| CC | 0.103 | 0.303 | 0.302 | −0.194 | 0.085 | −0.146 | −0.043 | | −0.073 | −0.118 |
| AS | −0.125 | 0.106 | −0.140 | −0.175 | 0.292 | −0.036 | 0.089 | −0.136 | | 0.278 |
| AA | 0.170 | 0.211 | −0.049 | −0.181 | −0.222 | 0.114 | −0.113 | −0.151 | 0.103 | |

\* Significant correlation at 0.05 level, ** significant correlation at 0.01 level; species codes are the same as in Table 1.

**Table 9.** Pearson (diagonal to the left) and Spearman's (diagonal to the right) correlation half-matrix table of main plant species in the humid habitats.

| | AP | CD | EC | DS | P | PT | C | CC | EP | CR |
|---|---|---|---|---|---|---|---|---|---|---|
| AP | | −0.530 ** | 0.146 | 0.003 | −0.306 | −0.127 | −0.099 | −0.132 | −0.411 * | −0.339 |
| CD | −0.505 ** | | −0.166 | 0.136 | 0.108 | 0.072 | −0.094 | 0.057 | −0.097 | 0.413 * |
| EC | 0.345 | −0.237 | | −0.245 | −0.238 | −0.456 * | −0.111 | −0.094 | 0.260 | −0.189 |
| DS | 0.220 | −0.030 | −0.221 | | −0.130 | −0.136 | −0.281 | 0.298 | 0.099 | 0.225 |
| P | −0.273 | 0.042 | −0.168 | −0.189 | | 0.319 | 0.139 | 0.018 | 0.180 | 0.073 |
| PT | −0.092 | −0.032 | −0.307 | −0.140 | 0.365 * | | 0.269 | −0.062 | −0.122 | −0.221 |
| Cd | 0.121 | −0.176 | −0.208 | −0.185 | 0.001 | 0.048 | | −0.048 | 0.385 * | −0.181 |
| CC | −0.148 | 0.126 | −0.166 | 0.076 | −0.069 | −0.088 | −0.080 | | 0.172 | 0.285 |
| EP | −0.399 * | −0.094 | 0.022 | −0.013 | 0.251 | −0.149 | 0.163 | 0.145 | | 0.084 |
| CR | −0.350 | 0.373 * | −0.230 | 0.077 | 0.003 | −0.199 | −0.160 | 0.261 | 0.048 | |

\* Significant correlation at 0.05 level, ** significant correlation at 0.01 level; species codes are the same as in Table 1.

The Spearman's rank correlation coefficient analysis (Tables 8 and 9) revealed that two species pairs were significantly positively correlated in the xeric environment. They were *C. dactylon*–*C. canadensis* and *S. viridis*–*A. subulatus*. Four species pairs, *A. philoxeroides*–*C. dactylon*, *A. philoxeroides*–*P. lapathifolia*, *E. crusgalli*–*P. thunbergia*, and *E. crusgalli*–*C. rotundus*, showed significant negative correlations. In the humid environment, two species pairs showed significant positive associations. They were *C. dactylon*–*C. rotundus* and *C. difformis*–*E. prostrata*. Three species pairs, *A. philoxeroides*–*C. dactylon*, *A. philoxeroides*–*E. prostrata*, and *A. philoxeroides*–*P. thunbergii*, showed a significant negative association.

Comparing the results of the $\chi^2$ test, Pearson correlation test, and Spearman's rank correlation test, we found that the significant rate of the test in xeric habitats was 18.42% (Spearman's rank correlation coefficient test) > 6.67% (Pearson correlation coefficient test) > 4.44% ($\chi^2$ test). The test significance ranking in humid environment was 11.11% (Spearman's rank correlation coefficient test) > 10.53% (Pearson correlation coefficient test) > 2.22% ($\chi^2$ test). The rates of significance observed using different tests were ranked in the same order, with the Spearman's rank correlation coefficient test showing a more significant association. This indicated that the Spearman's rank correlation coefficient test had higher sensitivity compared to the $\chi^2$ test and Pearson correlation coefficient test, and can reflect interspecific relationships more deeply and quantitatively accurately.

## 4. Discussion

In terms of the number of species, there was no significant difference between the dominant species in these two different environments (i.e., xeric vs. humid). The number of families and genera in the humid environment, however, was lower than in the xeric environment. One of the key explanations was that the sufficient water resource in the

humid environment caused a higher level of tillering rate of *A. philoxeroides* [43], which could rapidly occupy the environmental resources and, therefore, inhibit the growth of other families of plants.

From the relationship between the niche width and importance value, the overall trend of both types of habitats in this study showed that the greater (smaller) the importance value of the major species, the greater (smaller) its niche width. This was consistent with the view of Lu et al. that, in general, the importance value and niche width show a positive correlation, and the higher the importance value, the greater the resource utilization capacity and environmental adaptability of the species [44]. Moreover, by comparing the overlap index of the two types of environments, it can be seen that although the overlap index of the xeric environment was generally slightly higher than that of the humid environment, most of them ranged from 0 to 0.5 (Figure 2b). This indicated that most species in the two types of habitats invaded by *A. philoxeroides* do not compete much with each other, and their needs for environmental resources partially differ from each other, which leads to the coexistence on condition that resources are abundant [45].

The combination of plant niche width and niche overlap can further reveal community dynamics, species dominance, and other characteristics [46]. Generally, the niche width of a species is positively correlated with interspecific niche overlap [9,47]. In this study, the niche width of *A. philoxeroides* was absolutely dominant in both xeric and humid environments, and to a certain degree, the niche overlap was also generated between *A. philoxeroides* and other major species. We suggested that as an invasive species, *A. philoxeroides* has wider environmental adaptations than most other species because of its strong reproductive characteristics, tolerance ability [48], and ability to establish a symbiotic relationship in a short time. In contrast, the niche width of *P. thunbergii* was small. More importantly, the overlap and similarity between *P. thunbergii* and other species were low, indicating that the resource requirements of *P. thunbergii* were different to the other species.

The association coefficient (*AC*) revealed that the positive association between species was stronger in humid environments than in xeric environments. Although the communities with more positive interspecific relationships were likely to be more mature and stable [49], the communities in both humid and xeric environments were still considered unstable. During our field investigation, the xeric environments showed an overall non-significant positive association, and the $\chi^2$ test significance rate was only 4.44%, indicating that although the species formed certain relationships, there was also a certain competition for resources such as light, water, and soil nutrients [50], and the process of resource depletion might be accelerated under the intensified invasion of *A. philoxeroides*. Combining the results of the association coefficient and the $\chi^2$ test, most species were more compatible in terms of resource use in the humid environment.

The species pair *A. philoxeroides*–*E. crusgalli* showed more positive associations, indicating that they were the main companion for each other in most environments. This may be related to the biological characteristics of *E. crusgalli* in terms of its strong adaptability, vigorous growth, and high seed yield. The positive association of *A. philoxeroides*–*E. crusgalli* was better than that of other species pairs, provided that statistical errors and environmental and biological factors were excluded [13,51]. However, the number of non-significant species pairs was greater than the number of significant species pairs, indicating that the community structure was still unstable. Consistent with the findings on *Spermacoce alata* [52] and *Ageratina Adenophora* [53], we assumed that the communities were in the early stage of succession, and the invasion of *A. philoxeroides* had led to competition for resources and caused the instability in the community [38,54]. The analysis of external disturbances also indicated that, collaborating with the natural factors, such as superior topography and climatic conditions, the anthropogenic factors, including accelerated urbanization and human activities, may increase the invasion opportunities of plants and disrupt the balance and homeostasis of the local plant communities [55]. These trends were also revealed by the $\chi^2$ test, Pearson correlation test, and Spearman's rank correlation test.

Ecological niche characteristics are closely related to the interspecific associations between species, and they can together reflect the trends of community succession [56]. More specifically, the positive interspecific associations represent the similarity or overlap of resources requirements, while the negative associations represent the exclusion and separation of resource requirements [57]. Here, *A. philoxeroides* was closely associated with *E. crusgalli*, *C. dactylon*, and *S. viridis* in the xeric environment, which indicated that these species have similar resource preferences and environmental adaptations to *A. philoxeroides*. Moreover, this may constitute a more important hint that these species could be applied to prevent the spread of *A. philoxeroides* as they have a relatively large niche width and resistance to the invasion of *A. philoxeroides* in the xeric environment. At the same time, planting could involve *P. lapathifolia* and *A. australis*, which have larger niche widths but a smaller overlap with *A. philoxeroides*, to prevent vicious competition between species. The selection of species to prevent the invasion of target species also needs to avoid the use of alien species. Therefore, the native species *P. lapathifolia* and *A. australis* in the present study were selected instead of *C. rotundus* and *C. canadensis*, which have greater ecological niche width and overlap with *A. philoxeroides*. To achieve ecological control of *A. philoxeroides* in the humid environment, we suggested using *E. crusgalli*, *C. dactylon*, and *P. lapathifolia* as the main species, coupled with *P. thunbergii* and *D. sanguinalis*, which had a smaller overlap with *A. philoxeroides* in terms of resource requirements.

In addition, the biological control of *A. philoxeroides* invasion in China should also consider the economic value because they usually take place in important agricultural regions, such as the Jianghan Plain, and the alternative control of *A. philoxeroides* invasion in farmland should be considered to control the spread of *A. philoxeroides* by using plants with economic or ecological value through occupancy crowding. For instance, Wang et al. found that the growth and spread of *A. philoxeroides* could be effectively controlled using *Humulus scandens* [58]; Wang et al. researched the parasitism of *A. philoxeroides* by *Cuscuta australis*, and showed that the former could effectively inhibit the normal growth of the latter [59]; Deng Lili et al. found that *Ipomoea batatas* could effectively inhibit the spread of *A. philoxeroides* in agricultural fields based on the principle of interspecific plant competition for substitution [60].

## 5. Conclusions

The ecological niche and association between dominant species pairs in xeric and humid habitats invaded by *A. philoxeroides* at the plain lake area were studied in this research.

In both types of habitats invaded by *A. philoxeroides*, the dominant species with comparatively high importance values and ecological niche widths were members of the Graminaceae such as *E. crusgalli* and *C. dactylon*, except for *A. philoxeroides*. Among the 45 species pairs in the xeric habitats, there was only one pair of species with a niche similarity value $C_{ik} \geq 0.50$ and four pairs with a niche overlap value $O_{ik} \geq 0.50$; in the humid habitats, there was one pair of species with a niche similarity value $C_{ik} \geq 0.50$ and three pairs of species with a niche overlap value $O_{ik} \geq 0.50$. This indicated that there were few differences in ecological niches between the dominant plants in the invasive communities of *A. philoxeroides* in the two habitat types, with less similarity between species and little competition.

Combined with the $\chi^2$ test, Pearson and Spearman's rank correlation coefficient test, the test significance rates were 4.44%, 6.67%, and 18.42% in the xeric habitats, and 2.22%, 10.53%, and 11.11% in the humid habitats, respectively. All three tests indicated that the interspecific association and stability of the two types of habitats invaded by *A. philoxeroides* were poor as well as ecologically fragile.

Therefore, we concluded that the niche difference between the dominant plants of the two types of habitats invaded by *A. philoxeroides* was not highly variable, and the interspecific correlation was weak. The community structure of the two types of habitat invaded by *A. philoxeroides* was still in an unstable development stage.



The interspecific association between the dominant species in the humid environment was more complex than the xeric environment. In terms of niche requirements and habitat suitability, *A. philoxeroides* was likely to be more compatible with *E. crusgalli* and *C. dactylon* in xeric environments, while in the humid environment, *A. philoxeroides* was more highly correlated with *E. crusgalli*, *C. dactylon*, and *P. lapathifolia*. Species in the family Graminaceae, such as *E. crusgalli* and *C. dactylon*, had a certain resistance to the invasion of *A. philoxeroides*, which could be considered as alternative plants for the control of *A. philoxeroides* invasion in the plain lake area and the restoration and management of habitats that have been invaded.

**Author Contributions:** Conceptualization, Q.N., Q.Z. and L.Z.; methodology, Q.N. and Q.Z.; validation, Q.N.; formal analysis, Q.N.; investigation, Q.N., Q.Z., D.Z. and L.Z.; resources, Z.L.; data curation, Q.N. and Q.Z.; writing—original draft preparation, Q.N. and Q.Z.; writing—review and editing, X.L.; visualization, Q.N. and L.Z.; supervision, Z.L. and L.Z.; project administration, Z.L. and L.Z.; funding acquisition, Z.L. and L.Z. All authors have read and agreed to the published version of the manuscript.

**Funding:** This research was funded by the Hubei Agricultural Environmental Protection Station's agricultural invasive species census project in Hubei Province (2020108002001007).

**Institutional Review Board Statement:** Not applicable.

**Data Availability Statement:** The data presented in this study are available on request from the authors.

**Conflicts of Interest:** The authors declare no conflict of interest.

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
