# Peer review of "Niche and Interspecific Association of the Dominant Species during the Invasion of Alternanthera philoxeroides in the Yangtze River Basin, China"

_agriculture, doi:10.3390/agriculture13030621_

Round 1

Reviewer 1 Report

The manuscript of Nan et al. presents results of interesting study. Generally, they concern the influence of invasive species, Alternanthera philoxeroides, on two types of habitats. The study fit in important problem due to ecological and conservation points of view. The manuscript has some scientific value and results have practical aspect. Nevertheless They belong to the scope of the journal and may be published after major revisions. Below are my comments and suggestions.

Introduction

This chapter well introduce to the study problem. Nevertheless, I suggest to include some more general information about biological invasion, for example, in which way invasive species  reduce biodiversity.

Materials and methods

Row 74. Is: “We investigated the lake area…”. Authors studied rather some scientific problem in this area, not this area.

Row 83. The brackets should be removed.

Row 83. The invasive is rather the species, not community. Therefore I suggest to change: community of invasive A. philoxeroides. The next question is whether this species really forms community in the phytosociological sense? If not, may be it would be better to use other terms, for example patch.

Row 88. Please explain, in which way abundance, cover and high were measured? Sometimes abundance and cover is the same. Which method was used - Braun -Blanquet scale or other Individuals of a given species have often different size – did authors measure 30, 100 plants and averaged values?

Row 92. Please add the space before bracket.

Row 90-91. Two types or communities or two types of habitats – please, check the whole text according to this suggestion.

Rows  90-91. From each type of habitats 10 species were chosen, and in total there were 13 species. Explain it. Explain the difference between top species and dominant species. In which way domination was determined?

Row 93. Again,  A. philoxeroides communities? The title of Table 1. should be changed and this Table should be rearranged. Please give these data in one column.

Row 99. “Sum of the various resources..” What is the resource? May be it would be better to write “values of various parameters”. Resource/parameter is for example abundance, cover or high?

Row 102. and the next. What indicate certain plant species? These of 13 dominant species?

Row 106. What is relative density, relative frequency? These terms are not explained.

Row 128. Total species or total number of species?

Authors should order and better explain terminology in the whole text.

Results

This chapter is too descriptive and should be shortened. Differences between two habitat types should be better accented.

Subchapter 3.1. Two habitat types should be better compared. Important values should be changed to importance values. The number of species was similar in both habitats – there were the same species?

Row 197. Please add the species name to Polygonum or write Polygonum sp.

Rows 196-198. Please add the spaces in species names.

Subchapter 3.2. Please add the spaces in species names and also in the farther parts of the manuscript.

The title of Table 2. Is important value. Is it correct?

It will be better to connect Table 3 and Table 4. The same indexes in to habitat types would be clearer to comparison. 

Fig 3-6. Brackets are not needed in the titles of figures.

Discussion

Authors should wider explain obtained results. May be a few words about biological properties of species, which facilitate their  spread.

Row 390. VS. should be vs.

Row 393. Rater or rate?

Row 394. Growth concerns plants/ individuals not families.

Row 459 and 462. Should be “Species from Polygonum genus”.

Rows 461-464. In which way these species may be used to prevent A. philoxeroides invasion.

Conclusions

Again, authors have some problems with terminology, for example: invasive communities, life adaptation, alien community vegetation

Reviewer 2 Report

The following comments are recommended for the improvement of the paper.

1.       The abstract need revision according to the objectives of the study.

2.       The scientific names (Generic name) should not abbreviate when the author is writing for the first time either in abstract or throughout the text in paper.

3.       In introduction some updated and relevant paper must be cite of good journals.

4.       In the end of introduction the objectives of the study should be cleared.

5.       The author has missed the references in methodology section. It is very important when you following the methods using by some else and then must put the reference of that study.

6.       The data presentation is good but need clarification by providing us the supporting files for IV values. To see how IV are calculated as per methodology shown in Methodology section.

7.       The author has mentioned that “We concluded that the niche difference between the dominant plants of the two 479 types of invasive communities was highly variable, the interspecific correlation was weak put valid reason in result and discussion and in conclusion section.

8.       Double check all the references and its journal format.

Round 2

Reviewer 1 Report

Authors considered all suggestions and comments. The manuscript was importantly improved. The present version of the manuscript may be published in the journal.